# Study of the Antimicrobial Potential of the *Arthrospira platensis*, *Planktothrix agardhii*, *Leptolyngbya* cf. *ectocarpi*, *Roholtiella mixta* nov., *Tetraselmis viridis*, and *Nanofrustulum shiloi* against Gram-Positive, Gram-Negative Bacteria, and Mycobacteria

**DOI:** 10.3390/md21090492

**Published:** 2023-09-14

**Authors:** Alexander Lykov, Alexei Salmin, Ruslan Gevorgiz, Svetlana Zheleznova, Lyubov Rachkovskaya, Maria Surovtseva, Olga Poveshchenko

**Affiliations:** 1Novosibirsk Tuberculosis Research Institute MH RF, Okhotskaya 81 A, Novosibirsk 630040, Russia; salmin.a@list.ru; 2Research Institute of Clinical and Experimental Lymphology—Filial of the Institute of Cytology and Genetics, Timakova 2, Novosibirsk 630060, Russia; noolit@niikel.ru (L.R.); mfelde@ngs.ru (M.S.); poveschenkoov@yandex.ru (O.P.); 3Kovalevsky Research Institute of Biology of Southern Seas RAS, Nakhimova 2, Sevastopol 299011, Russia; r.gevorgiz@yandex.ru (R.G.); zheleznovasveta@yandex.ru (S.Z.)

**Keywords:** diatoms, cyanobacter, green microalgae, gram-positive and gram-negative bacteria, mycobacteria, antimicrobial potential

## Abstract

The incidence of diseases brought on by resistant strains of micro-organisms, including tuberculosis, is rising globally as a result of the rapid rise in pathogenic micro-organism resistance to antimicrobial treatments. Secondary metabolites with potential for antibacterial activity are produced by cyanobacteria and microalgae. In this study, gram-positive (*S. aureus*, *E. faecalis*) and gram-negative (*K. pneumoniae*, *A. baumannii*, *P. aeruginosa*) bacteria were isolated from pulmonary tuberculosis patients receiving long-term antituberculosis therapy. The antimicrobial potential of extracts from the cyanobacteria *Leptolyngbya* cf. *ectocarpi*, *Planktothrix agardhii*, *Arthrospira platensis*, *Rohotiella mixta* sp. nov., *Nanofrustulum shiloi*, and *Tetraselmis (Platymonas) viridis* Rouchijajnen was evaluated. On mouse splenocytes and peritoneal macrophages, extracts of cyanobacteria and microalgae had inhibitory effects. In vitro studies have shown that cyanobacteria and microalgae extracts suppress the growth of bacteria and mycobacteria. At the same time, it has been demonstrated that cyanobacterial and microalgal extracts can encourage bacterial growth in a test tube. Additionally, the enhanced fucoxanthin fraction significantly reduced the development of bacteria in vitro. In a mouse experiment to simulate tuberculosis, the mycobacterial load in internal organs was considerably decreased by fucoxanthin. According to the information gathered, cyanobacteria and microalgae are potential sources of antibacterial compounds that can be used in the manufacturing of pharmaceutical raw materials.

## 1. Introduction

With their vast diversity and capacity for habitat adaptation, cyanobacteria and microalgae are rich sources of biologically active substances, such as primary (proteins, lipids, and polysaccharides) and secondary (phycocyanin, oleic acid, palmitoleic acid, beta-carotene, lutein, and zeaxanthin) metabolites [1,2,3]. The type of lighting used for cultures in bioreactors can control the biosynthesis of secondary metabolites, with red light stimulating the synthesis of polyphenols and green light stimulating the synthesis of flavanoids [4,5]. Since harmful compounds can be produced by cyanobacteria and microalgae, only a small selection of microalgae can be used as a source of organic food, feed, nutraceuticals, and pharmaceuticals. Therefore, *Chlorella vulgaris*, *Dunaliella*, *Aphanizomenon*, and *Nostoc* are permitted for human consumption [6,7,8]. The issue is that few of the examined cyanobacteria and microalgae have GRAS status, which restricts their application as nutraceuticals. It has been established that members of the green, diatom, and golden algae classes primarily exhibit antimicrobial activity, and that water pollution has an impact on the level of antimicrobial synthesis [9]. It has been proven that extracts of several microalgae have antibacterial properties. For instance, *Arthrospira platensis* extract inhibited *Staphylococcus haemolyticus*, *Streptococcus mutans*, and *Bacillus subtilis* strains; *Cladophora rupestris*, *Furcellaria lumbricalis,* and *Ulva intestinalis* inhibited *Streptococcus mutans* strains [10]. Therefore, sulfoquinovosyldiacylglycerol, monogalactosylmonoacylglycerol, hexadeca-4,7,10,13-tetraenoic acid, palmitoleic acid, and lysophosphatidylcholine obtained from cyanobacteria and microalgae have antimicrobial and antifouling properties for microbes [11]. In vitro, *Scenedesmus obliquus* extract in dichloromethane enhanced the antibacterial activity of penicillin and fluoroquinolones against *Staphylococcus aureus*, *Escherichia coli*, and *Salmonella typhimurium* strains [12]. *Chlorella vulgaris* [13], *Skeletonema tropicum* and *Chaetoceros Pseudocurvisetus* [14], and *Chaetoceros muelleri* [15] extract have antifungal activity, as do the fatty acids (oleic, linoleic, and linolenic acids) and carotenoids (xanthin, neoxanthin, cryptoxanthin, and echinenone) in those plants. Few research studies have compared the antimicrobial effects of cyanobacteria and microalgae extracts from various taxa and habitats, whose biomass we were able to get in industrial photobioreactors, notwithstanding the studies that have been provided above. This is significant because it is generally accepted that high-value products that cannot be grown on an industrial scale are of little interest.

## 2. Results

### 2.1. Cyanobacteria and Microalgae’s Biochemical Profiles

In the dry biomass of various cyanobacteria and microalgae species, proteins, lipids, and carbohydrates were identified (Table 1). The lowest levels of protein were found in *N. shiloi* and *T. viridis*; the lowest levels of lipids were found in *R. mixta* sp. nov. and *P. agardhii*; and the highest levels of carbohydrates were found in *T. viridis* and R. *mixta* sp. nov.

Only in *N. shiloi* was FX found. *L. ectocarpi*, *R. mixta* sp. nov., and *P. agardhii* all had their phytobiliproteins identified. Additionally, the PUFA concentration of cyanobacteria and microalgae varies.

Proteins (50–70%), lipids (5%), carbohydrates (10–20%), pigments, amino acids, fatty acids, and PUFA are among the well-studied biochemical substances in *A. platensis* [16]. Additionally, the fermentative analysis of the chemical makeup of the bioactive substances in cyanobacteria and microalgae extracts revealed the presence of proteins, lipids, and glucose in these extracts (Figure 1).

### 2.2. The Ceanobacteria and Microalgae Extract’s Cytotoxic Potential

On peritoneal macrophages and splenocytes of Balb/c mice, the MTT assay was performed using aqueous extracts of the various cyanobacteria and microalgae species at a concentration of 1% *v*/*v*. Cyanobacteria and microalgae extract were shown to have a significant amount of cytotoxic activity (up to 80%) when tested on splenocytes (Figure 2). Mice’s peritoneal macrophages showed both a decline and an increase in metabolic activity (NADPH-dependent oxidoreductase).

### 2.3. Cyanobacteria and Microalgae Extracts Have Antibacterial Properties

We first evaluated diatom extracts with a high FX concentration (*A. platensis* and *N. shiloi*) to determine the antibacterial activity of cyanobacteria and microalgae (Table 2). The bacteria in the nutritional medium were given 1% *v*/*v* of the test extract for this purpose, and growth was assessed after 14 and 21 days. On days 14 and 21, in particular, FX from *A. platensis* and *N. shiloi* inhibited the growth of gram-positive (*S. aureus*, *E. faecalis*) and gram-negative (*K. pneumoniae*, *A. baumannii*, and *P. aeruginosa*) bacteria. On day 21, FX’s stability and degradation can cause its antibacterial capability to decline.

Then, using ceftazidime and FX as a comparison, we obtained DMSO (1%) extract from cyanobacteria and microalgae and examined their antibacterial potential at 1% *v*/*v*, 0.5% *v*/*v*, and 0.25% *v*/*v* in vitro (Table 3). Cyanobacteria and microalgae extracts did not appear to have any influence on bacterial growth in vitro that was dose-dependent. The growth of bacteria was inhibited by FX in a dose-dependent manner. In addition, certain cyanobacteria and microalgae extracts have been shown to promote bacteria growth when compared to ceftazidime and controls (*p* < 0.05).

A wide variety of bioactive compounds, such as carotenoid and chlorophylls compounds with antimicrobial activity, were synthesized by cyanobacteria. On the growth inhibition of bacterial strains resistant to antimicrobial medicines in vitro, we compared the effects of cyanobacteria and microalgae extract, chlorophyllipt (a combination of chlorophylls from eucalyptus leaves), and other compounds (Table 4). In contrast to chlorophyllipt and ceftazidime, we discovered that FX efficiently suppressed bacterial growth (*p* < 0.05).

The results of multiple in vitro experiments examining the effects of enhanced FX fraction, cyanobacteria and microalgae extracts, and comparative analysis with chlorophyllipt all point to the presence of physiologically active compounds with antibacterial activity in the extract. Given that cyanobacteria and microalgae extracts contain a wide variety of nutrients (proteins, fats, and sugars), we have demonstrated through experiments that this affects the antibacterial potential, helping bacteria survive in the presence of high extract concentrations in the culture medium rather than being killed off.

### 2.4. Microalgae’s Antimycobacterial Potential

We employed two *Mycobacteriaceae*—*Mycobacterium tuberculosis* virulent type strain H37Rv and *Mycobacterium smegmatis*, a nonpathogenic microbe characterized by rapid growth ability—to establish the antimycobacterial activity of cyanobacteria and microalgae extract.

First, we compared rifampicin’s antimycobacterial activity to that of *A. platensis* aqueous extract and FX (Figure 3). Only FX significantly (*p* < 0.05) reduced the growth of *Mycobacterium smegmatis* and *Mycobacterium tuberculosis*.

The antimycobacterial activity of cyanobacteria and microalgae in the DMSO extract was then compared to that of chlorophyllipt and rifampicin (Figure 4; *p* < 0.05). Extracts from *N. shiloi, L. ectocarpi*, and FX from *N. shiloi* inhibit mycobacteria growth more effectively. Some extracts of cyanobacteria and microalgae promoted mycobacterial development, while others stifled it. Such a distribution of the effects of cyanobacteria and microalgae extract may indicate the presence of efflux pumps and the resistance of pathogenic mycobacteria strains to the action of toxic substances. The presence of nutrients in cyanobacteria and microalgae extract may have the unintended consequence of stimulating the growth of mycobacteria.

Additionally, it made sense to determine the dose-dependent impact of cyanobacteria, microalgae, and FX on the persistence of mycobacteria in vitro (Table 5).

The growth of mycobacteria has been demonstrated to be either inhibited or stimulated by cyanobacteria and microalgae extracts, and there is no obvious dose dependence. The minimum or maximum concentration of cyanobacteria and microalgae extract occasionally encouraged or inhibited mycobacteria growth.

### 2.5. Effect of Fucoxanthin Administration in Tuberculosis-Infected Mice

The outcome of oral FX treatment was then investigated in mice with experimental tuberculosis brought on by the injection of Mycobacterium tuberculosis H37Rv into the tail vein. When compared to the untreated group and the rifampicin-treated group, the administration of FX decreased the number of CFU of *Mycobacterium tuberculosis* growth on solid medium from the lungs by 46% and 8%, respectively (Figure 5; *p* < 0.05). When compared to untreated mice, rifampicin and FX have no effect on *Mycobacterium tuberculosis* in the liver (*p* > 0.05). When compared to the untreated group and the rifampicin-treated group, FX administration has the ability to reduce the number of CFU of *Mycobacterium tuberculosis* growth on solid medium from the spleen by 30% and 23%, respectively (*p* < 0.05).

## 3. Discussion

In this work, we evaluated the biochemical components and antibacterial potential of algae from various taxa (diatoms, cyanobacteria, and green microalgae), environments (seas, rivers, and lakes), and microbial communities (G-positive and G-negative). Aside from *A. platensis*, we also used cyanobacteria and microalgae (*L. ectocarpi*, *N. shiloi*, *T. viridis*, *P. agardhii*, and *R. mixta* sp. nov.) that have not been investigated for their antibacterial potential. As a result, *L. ectocarpi* microalgae metabolites such as proteins, carotenoids, phenols, and chlorophyll have the power to improve melatonin production, decrease the negative effects of free radicals, and inhibit the activity of matrix metalloproteases, such as elastase and hyaluronidase. A study on the manifestation of antibacterial activity of an *L.* sp. extract against MSSA and MRSA exclusively followed the extract’s preceding laser treatment [17,18,19]. It has been demonstrated that the cyanobacterium *P. agardhii* exhibits estrogenic activity [20]. The secondary metabolites carotene, zeaxanthin, 3-hydro-carotene, 3-hydroxyechinenone (4-keto-3′-hydroxyl-β-carotene), echinenone, caloxanthin, β-cryptoxanthinoleate, and antheraxanthin were discovered in the cyanobacterium *R. mixta* sp. nov. only in 2021 [21]. Except for the enriched fucoxanthin fraction from *A. platensis* and *N. shiloi* produced by extraction in ethanol and then in olive oil, we employed unfractionated cyanobacteria and microalgae extracts. The identification of all chemicals is illogical within the parameters of the task outlined in our work, which involves a thorough examination of several species of cyanobacteria and microalgae.

A valuable source of nutrients, such as proteins, lipids, and carbohydrates, is cyanobacteria and microalgae. The type of illumination of cultures in bioreactors, where red light stimulates the synthesis of polyphenols and green light—flavanoids—the rate of aeration, and the composition of nutrient media can all affect the process of photosynthesis of bioactive substances in cyanobacteria and microalgae [4,22].

The primary metabolites (proteins, carbohydrates, and lipids, including tiglycerides) as well as fucoxanthin, a pigment from the group of carotenoids, were found in the metabolites of cyanobacteria and microalgae that were collected for investigation. Phycobiliproteins (phycoerythrin and phycocyanin) and mono- and polyunsaturated fatty acids (-linolenic acid, linoleic acid, oleic acid, and -linolenic acid) were also found in the composition of cyanobacteria and microalgae. Our findings about the biochemical makeup of cyanobacteria and microalgae are consistent with the findings of other researchers, who point out that these organisms are a rich source of nutrients and biologically active chemicals [4,22].

We determined how cytotoxic cyanobacterial and microalgae extracts were to somatic cells and microbes. We have demonstrated the cytotoxicity of cyanobacterial and microalgae extracts against mouse immune cells (peritoneal macrophages and spleen lymphocytes), which is also consistent with the results of other researchers who have examined the presence of cytotoxicity of different cyanobacteria and microalgae against somatic cells, including tumor cells [20].

Searching for new sources of biologically active molecules with antibacterial activity is important due to the rise in the incidence of microbes resistant to the majority of the antimicrobial drugs currently in use. With their capacity to fend off and defend against diseases, cyanobacteria and microalgae are valuable sources of such compounds [23]. Alkaloids, fatty acids, indoles, macrolides, peptides, phenols, pigments, and terpenes are some of the antibacterial substances created by cyanobacteria [24,25]. The kind of algae, the extractor, and the concentration of the extract all affect the antibacterial potential of cyanobacteria, macro-, and microalgae [26].

The bioactive molecules from cyanobacteria and microalgae are extracted using a variety of solvents, including hexane, chloroform, methylene chloride, ethyl acetate, methanol, acetone, and water [15]. In our study, the eluents used were distilled water, 1% DMSO solution, and 95% ethanol. We noted the antibacterial potential of water- and DMSO-based extracts of cyanobacteria and microalgae from the in vitro data. We also discovered fucoxanthin’s antibacterial properties after extracting it from ethanol. Our findings are consistent with the findings of the authors’ studies, which showed that gram-positive and gram-negative bacteria were inhibited from growing when exposed to an alcoholic extract of *Chlorella* sp. UKM8 contained the antimicrobial compounds phenol, hexadecanoic acid, phytol, 9, 12-octadecadienoic acid, and bicyclo [3.1.1] heptanes [27]. *Scenedesmus subspicatus* is used as an example to demonstrate the feasibility of utilizing several types of eluents with different polarities (ethanol, methanol, butanol, acetone, DMSO, and water) and assessing the antibacterial ability of the extracts [28]. Cyanobacterial and microalgae extracts had increased antioxidant and antibacterial activity when used with water and DMSO as eluents. When employed as an eluent, acids can replicate their antibacterial properties, as demonstrated in [23].

The minimum inhibitory concentration method and its modification as a micromethod, which are simple to set up and reproduce and are practical for meta-analyses, are two ways to evaluate the inhibitory impact of bioactive substances [29]. The MIC of the cyanobacteria and microalgae extracts was also determined using the micromethod. The obtained data on MIC of our extracts are comparable to the data of other researchers who found that extracts of *Isochrysis galbana, Scenedesmus* sp. NT8c, *Chlorella* sp. FN1, *Ettlia pseudoalveolaris* [30,31], *Oscillatoriales*, *Nostocales* [25], *Amphidinium carterae* [32,33], *Chlorella vulgaris,* and *A. platensis* [34,35] inhibited the growth of gram-positive and gram-negative bacteria. Linoleic, oleic, docosahexaenoic, eicosapentaenoic acids, polyphenols, amphidinolides, lipid complexes, and nanopeptides (cyclic peptides) are responsible for the antibacterial properties of extracts from cyanobacteria and microalgae.

If the cyanobacteria and microalgae we tested had the chemical makeup described above, it may be inferred that the majority of these bioactive compounds are also present in the extracts produced after extraction using eluents (water, DMSO, and ethanol). It will take more research to confirm them.

Despite the successful battle against tuberculosis, antibiotic-resistant mycobacteria are becoming more common, necessitating the search for novel substances with antituberculosis action, including those from natural sources.

According to the findings of our in vitro and in vivo research, most cyanobacterial and microalgae extracts have strong antituberculosis efficacy, particularly in the enhanced fucoxanthin fraction. Our findings are in agreement with studies that have also revealed the antituberculosis effect of cyanobacteria and microalgae, in particular for *Chlorella vulgaris* [13], *Skeletonema tropicum,* and *Chaetoceros pseudocurvisetus* [14], which is due to the combination of unsaturated fatty acids (oleic, linoleic, and linolenic acids) and carotenoids (xanthin, neoxanthin, cryptoxanthin, and echininone). The antituberculosis activity of extracts from *N. tuberculosis*, *P. agardhii*, and *L. ectocarpi* against various mycobacterial strains has also been revealed for the first time, and the presence of fucoxanthin has been confirmed not only under MIC data but also on the mouse model of tuberculosis infection. Fucoxanthin’s ability to prevent tuberculosis is based on its ability to block the enzymes UDP-galactopyranose mutase (UGM) and arylamino-N-acetyltransferase, which are essential for the formation of the mycobacterial cell wall [36].

It should be noted that the antimicrobial activity of cyanobacteria and microalgae extracts is not always detected. This is most likely because there are a lot of extracted primary and secondary metabolites that can both inhibit and stimulate the growth of micro-organisms, given the presence of antioxidant potential. This is consistent with the data on bacterial symbiosis with cyanobacteria and microalgae [37,38,39,40].

This study’s findings demonstrated that nonfractioned extracts and fucoxanthin from various cyanobacteria and microalgae taxa were efficient against gram-positive and gram-negative bacteria as well as mycobacteria (*Mycobacterium tuberculosis* strain H37Rv and *Mycobacterium smegmatis*). It must be highlighted, nevertheless, that some cyanobacteria and microalgae promoted the growth of bacteria.

The utilization of unprocessed fractions of cyanobacteria and microalgae extract is one of the article’s limitations. These investigations, which will determine the most significant components of cyanobacteria and microalgae extracts with antibacterial activity, are planned for the future.

## 4. Materials and Methods

### 4.1. Microalgae

The study included cyanobacteria and microalgae from a variety of systematic groups that were collected from the Kovalevsky Research Institute of Biology of the Southern Seas RAS’ Collective Use Center “Collection of Hydrobionts of the World Ocean.” A.A. Goncharov, Vladivostok, donated the cyanobacterium *R. mixta* sp. nov. as the sole exception. Marine cyanobacteria (*L. ectocarpi—Leptolyngbya* cf. *ectocarpi*), diatoms (*N. Nanofrustulum*—*Nanofrustulum shiloi*; GenBank OR359397) and green microalgae (*T.* viridis—*Tetraselmis (Platymonas) viridis* Rouchijajnen), freshwater cyanobacteria (*P. agardhii —Planktothrix agardhii* and *A. platensis—Arthrospira platensis*; https://www.ncbi.nlm.gov/nuccore/MZ408912.1 (accessed on 20 August 2023)), and soil cyanobacteria (*R. mixta* sp. nov.–*Roholtiella mixta* sp. nov.; https://www.ncbi.nlm.nih.gov/nuccore/MK990636.1 (accessed on 20 August 2023)) were included in the study.

*L. ectocarpi* and *P. agardhii* cyanobacteria were discovered in 2020 from periphyton in Karantinnaya Bay in the Black Sea (44°36′56″ N, 33°30′10″ E) following the exposing of solid substrates. A sample with a biofilm covering an area of 1 cm^2^ was placed in a Petri dish with 30 mL of the liquid growth medium BG-11 prepared on sterile seawater (g/L) to produce a batch culture of periphyton cyanobacteria: NaNO_3_ = 1.5, K_2_HPO_4_ = 0.04, MgSO_4_ = 0.075, CaCl_2_ = 0.036, citric acid = 0.006, ferric citrate = 0.006, Na_2_EDTA = 0.01, and Na_2_CO_3_ = 0.02. The culture was kept at a temperature of 23 ± 2 °C under natural light for 2–4 weeks until noticeable signals of development appeared. These samples were located using a guide.

The cultures were acclimated to the experimental conditions in the initial phase of the study. In 1 L flasks, the inoculum was produced using the accumulation mode. On mineral nutrition media made with sterilized seawater and distilled water, cultures were cultivated.

The BG11 medium was used to cultivate *N. shiloi*, *L. ectocarpi*, and *R. mixta* sp. nov., Trenkenshu media for *T. viridis*, and Zarruk medium with some modifications for *P. agardhii* and *A. platensis*.

### 4.2. Chemical Analysis

Ten grams of wet cyanobacteria and microalgae biomass were treated with a chlorophorm-ethanol mixture (2:1) to extract the lipids until the biomass turned fully discolored. To eliminate nonlipid impurities, the extract was washed with water three to four times. The gravimetric methods were used to calculate the amount of total lipids present in the chlorophorm fraction [41]. FAs were hydrolyzed and methylated in order to assess the content of fatty acids (FAs) in the total lipid extracts and lipid fractions. On a rotary evaporator, the chlorophorm fraction was evaporated, and the residue was mixed with 5 mL of freshly made alkali solution in methanol (10 mL 3 N NaOH and 90 mL 90% methanol). The resultant mixture was then cooked under reflux for 1.5 h to achieve complete saponification. The mixture was then strengthened with a few drops of a 1% alcoholic phenolphthalein solution, decontaminated with three pieces of 5 mL of hexane, neutralized with a few drops of 0.1 N HCl, and then re-extracted in two to three batches using portions of 5 mL of hexane. The fractions of hexane were gathered and mixed. At a temperature of 30 °C, the hexane solution was evaporated to dryness on a rotary evaporator. A 5-mL, 3% hydrogen chloride in methanol solution was added to the residue in order to esterify the FAs. The resultant solution was reflux-boiled for two hours, cooled, and then extracted with hexane (3–5 mL). Layers of hexane were mixed. The hexane fraction was kept at a temperature of 20 °C for no more than one day prior to the detection of fatty acid methyl esters (FAME). The resultant FAs methyl esters (FAMEs) were examined by gas chromatography utilizing a Chromatec Crystal 5000.2 apparatus (SGE Analytical Science, Kiln Farm Milton Keynes, United Kingdom) that was outfitted with an MS detector and a capillary BPX5 column (60 m, 0.25 mm, and 0.25 m). A split/splitless injector and a flame-ionization detector were also used by the apparatus, both of which worked at 280 °C. The NIST 14 library was used for the data analysis. FAMEs were recognized using the Supelco FAME 10 mix 37 (Hangzhou Keyinchem CO., LTD, Hangzhou, China) approved standard. The number of fatty acids was calculated as mg per gram of dry weight.

The Lowry methods were used to determine the protein content of cyanobacteria and microalgae biomass [42]. Dubois methods were used to determine whether there were any sugars in the cyanobacteria and microalgae biomass [43].

According to [44], the FX concentration in the cyanobacteria and microalgae biomass was calculated. At a wavelength of 488 nm, the extinction coefficient was calculated to be 1280 mL/g·cm [45]. The equations provided by Bennet and Bogorad [46] were used to calculate the contents of C-phycocyanin and C-phycoerythrin.

Protein levels were assessed in DMSO-extracts of cyanobacteria and microalgae using a photometric approach with bromocresol green, triglyceride levels using an enzymatic colorimetric method called GPO-PAP, and glucose levels using a glucose oxidase method called GOD-PAP.

### 4.3. Fucoxanthin Extraction

The suspension was centrifuged at 1600× *g* for 15 min to determine the *Nanofrustulum shiloi’s* rough mass. It was eliminated by the supernatant. Fucoxanthin (FX) was extracted from 100 g of crude *Nanofrustulum shiloi* biomass at 38–40 °C using a minimum volume of 96% ethanol (200 mL). In these circumstances, a 2-h extraction of FX from the biomass that was nearly complete (90–95%) was noted. An alcoholic extract with a final concentration of 0.35 mg/mL FX was produced as a result.

The high FX concentration alcohol extract was evaporated on a rotary evaporator at 1 kPa and 40 °C to a minimum volume (5–7) mL, and 200 mL of olive oil was then added to create FX-enriched oil. After that, more of this combination was evaporated in order to completely remove the ethanol from the olive oil. This led to the creation of an oil-based FX solution with a 0.5 mg/mL concentration.

### 4.4. Microalgae Extracts

By drying a 3–5 mm layer of biomass on polyethylene placed on a flat surface in a warm air current (38 °C) to a residual humidity of 9–10%, the dry mass of several cyanobacteria and microalgae species was obtained. The dried mass of microalgae was kept at 18 °C in a container that was tightly sealed. From 1 g of biomass, bioactive compounds were extracted from cyanobacteria and microalgae using a minimal volume of distilled water or a 1% DMSO solution (10 mL) at 37 °C for 24 h. After filtering and precipitating the compounds with a 14,000× *g* centrifuge, the supernatants were used for in vitro research.

### 4.5. Cytotoxicity Assay

All experimental methods were carried out in conformity with regional institutional ethics committee approval and national and EU regulations for animal experimentation (protocol No. 56).

Mouse Balb/c peritoneal macrophages and splenocytes were tested for viability using the MTT (3-(4, 5-dimethylthiazol-2-yl)-2, 5-diphenyl trazolium bromide) assay. Splenocytes were acquired via spleen homogenization, while macrophages were collected from peritoneal lavage. Peritoneal macrophages or splenocytes at 10^5^/well in RPMI-1640 culture medium supplemented with 10% FCS, 2 mM L-glutamine, and 100 U/mL antibiotic/antimycotic were incubated with or without 1% *v*/*v* of various microalgae extracts for 24 h at 37 °C in a humid environment with 5% CO_2_.

### 4.6. Antimicrobial Assay

A collection of bacteria, including the museum strains MSSA ATCC 25923 and MRSA BAA-1690 and clinical isolates from patients with multidrug-resistant tuberculosis infection, including *Pseudomonas aeruginosa* (*P. aeruginosa*) resistant strain from surgical materials isolated and sensitive strain sputum isolated, *Staphylococcus aureus* (*S. aureus*) sputum isolated, *Klebsiella pneumoniae* (*K. pneumoniae*) sensitive strain and resistant from sputum or urea isolated (respectively); *Streptococcus pyogenes* (*S. pyogenes*) sensitive strain urea isolated, *Acinetobacter baumannii (A. baumannii*) resistant strain sputum isolated, *Enterococcus faecalis* (*E. faecalis*) resistant strain urea isolated, and different mycobacteria (*Mycobacterium tuberculosis* strain H_37_Rv and *Mycobacterium smegmatis*) were used for the MIC (minimum inhibitory concentration) assay.

All clinical isolates of bacteria were collected from Novosibirsk Tuberculosis Institute patients who had pulmonary tuberculosis and are kept in a pathogen bank. The MALDI-TOF approach was used using a Microflex mass spectrometer (Bruker Daltonics, GmbH&Co. KG, Bremen, Germany) to identify micro-organisms. In accordance with the recommendations of the Clinical and Laboratory Standards Institute (CLSI 2012), the disc diffusion method was used to assess the susceptibility of bacteria to antibiotics.

After being exposed to extracts from several microalgae taxa for 24 h, 14 days, and 21 days at concentrations of 1, 0.5, and 0.25% *v*/*v* with 35, 17.5, and 8.75 g/mL FX, the antibacterial activity was assessed. Ceftazidime (50 g/mL), rifampicin (100 g/mL), and culture media RPMI 1640 with 10% FCS were utilized as control samples. On sterile 96- or 48-well plastic plates, cyanobacteria and microalgae extract MIC tests were conducted. Briefly, an inoculum of microbes (approximately 1.5 × 10^7^ CFU/mL) in 90 µL and different concentrations of microalgae extracts (1% v/v, 0.5% *v*/*v,* and 0.25% *v*/*v*) in 10 µL were added to the wells for 96-well plates, and an inoculum of microbes in 0.9 mL, and cyanobacteria and microalgae extracts in 100 µL for 48-well plates were incubated for 24 h in a humidified state with 5% CO_2_ at 37 °C. In triplicate, each extract underwent treatment. On the following days, days 14, and 21, 10 L of MTT were added to each well, followed by the addition of DMSO 4 h later to dissolve the formazan. At 570 nm, the absorbance was then measured.

### 4.7. Mice Model of Tuberculosis

A total of 15 Balb/c mice aged 8–12 weeks received an intravenous infection with 5 × 10^6^ CFU of the *Mycobacterium tuberculosis* strain H37Rv. The mice were divided into three groups of five mice each on day 30: *Mycobacterium tuberculosis* strain H37Rv-injected mice were left untreated (TB-untreated); Rifampicin 0.17 g/mice day in 0.5 mL of 0.9% NaCl solution was administered per os over the course of five days, once per day; and Fx, 87.5 mg/mL of oil extract, was given per os over the course of five days, once per day. Animals were killed by cervical dislocation on day 8 after the start of the treatment, and their lungs, liver, and spleen were removed. After homogenizing the samples, BD BBL^TM^ MycoPrep^TM^ Reagent (BD, San Jose, USA), phosphate buffer saline washing, and Lowenstein–Jensen solid medium plating, the results were examined. On day 21, CFU counts were calculated.

### 4.8. Statistical Analysis

Utilizing Statistica 10.0 for Windows, data was examined. The data was presented in tables as mean ± standard deviation (SD). The data was analyzed using one-way analysis of variance (ANOVA) with a Bonferroni correction (Bonferroni post hoc test) to examine differences between groups. In this study, the normality of the distribution was assessed using the Shapiro–Wilkes criterion. *p*-values of 0.05 or less were regarded as statistically significant.

## 5. Conclusions

In this study, we evaluated the antibacterial activity of cyanobacterial and microalgae extracts against bacteria isolated from various sources, including individuals with MDR tuberculosis and *Mycobacteria* strains. Our research findings showed that in vitro cytotoxicity against mouse immune cells, gram-positive and gram-negative bacteria, and *mycobacteria* was produced by cyanobacterial and microalgal water or DMSO extracts and ethanol extracts of fucoxanthin. Fucoxanthin oil extract taken orally reduced the burden of mycobacteria in internal organs in mice with tuberculosis. These compelling findings suggest that cyanobacterial and microalgae extracts could be suitable candidates for the creation of antibacterial drugs.

## Figures and Tables

**Figure 1 marinedrugs-21-00492-f001:**
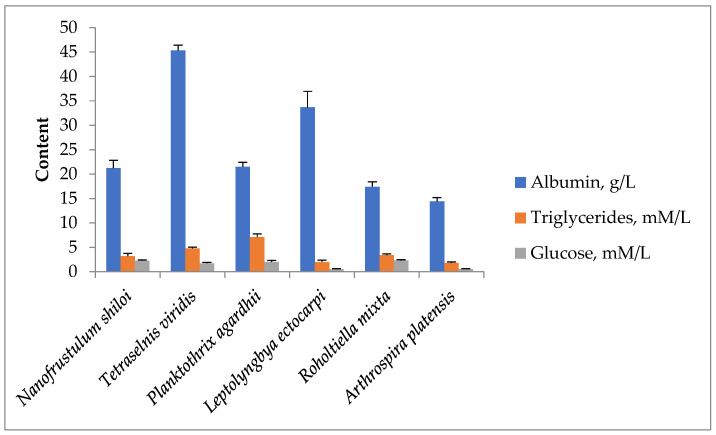
Content of the primary biochemical substances in the various cyanobacteria and microalgae taxa’s DMSO extracts.

**Figure 2 marinedrugs-21-00492-f002:**
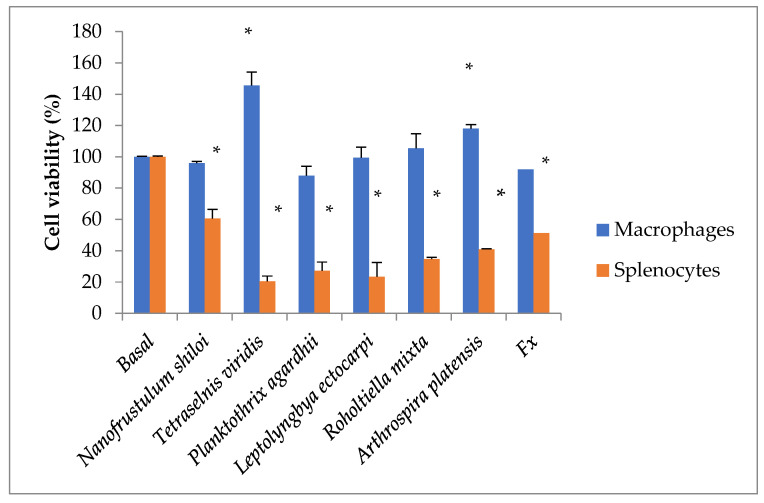
Cell viability of the Balb/c mice’s peritoneal macrophages and splenocytes after a 24-h exposure period with various cyanobacteria and microalgae water-extract by MTT assay. FX, fucoxanthin from *Nanofrustulum shiloi.* Statistical notations: * *p* < 0.05.

**Figure 3 marinedrugs-21-00492-f003:**
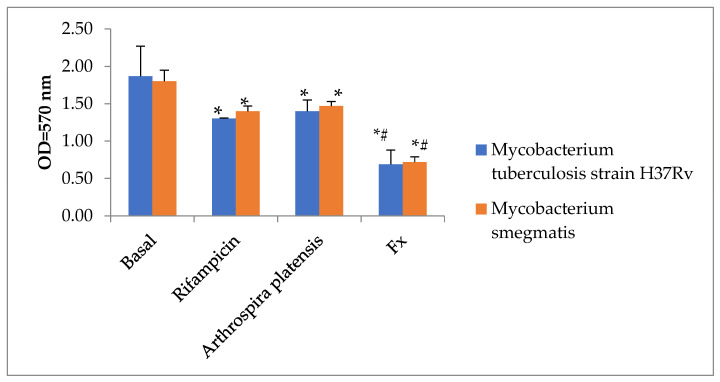
Fucoxanthin and *Arthrospira platensis* water extract both have antimycobacterial properties. Statistical notations: * = *p* < 0.05 compared with basal mycobacteria growth; # = *p* < 0.05 with rifampicin. Rifampicin was added to wells in doses of 100 µg/mL, *A. platensis* in doses of 1% *v*/*v,* and FX in doses of 35 µg/mL.

**Figure 4 marinedrugs-21-00492-f004:**
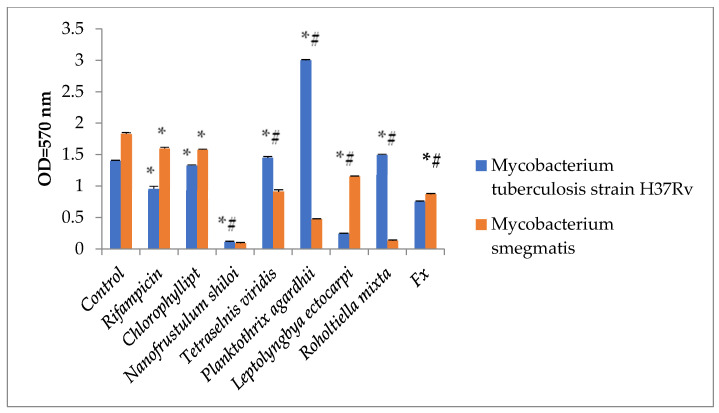
Comparison of the antimycobacterial properties of DMSO from microalgae and chlorophyllipt extracts (M ± SD). Statistical notations: * = *p* < 0.05 compared with basal mycobacteria growth; # = *p* < 0.05 with rifampicin. Rifampicin was added in wells in doses of 100 µg/mL, chlorophyllipt and microalgae extracts in doses of 1% *v*/*v,* and FX in doses of 35 µg/mL.

**Figure 5 marinedrugs-21-00492-f005:**
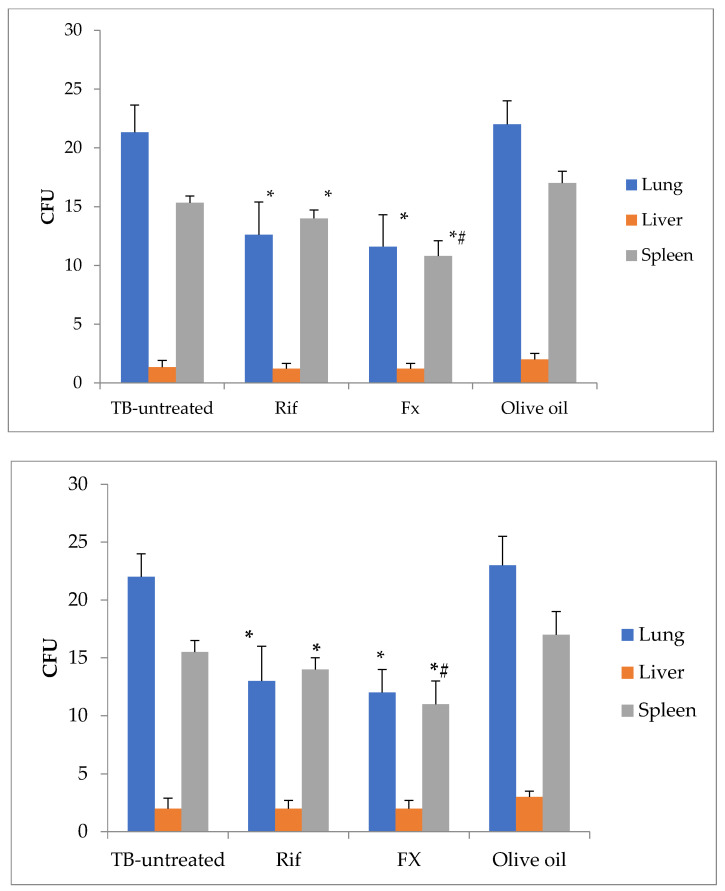
The amount of *Mycobacterium tuberculosis* CFU growth on Lowenstein–Jensen solid medium from the liver, spleen, and lungs of tuberculosis-infected mice (M ± SD). CFU, colony-forming units; Rif, rifampicin-treated group; FX, fucoxanthin-treated group. Statistical notations: * = *p* < 0.05 compared with TB-untreated group; # = *p* < 0.05 with rifampicin-treated group.

**Table 1 marinedrugs-21-00492-t001:** Metabolites of Cyanobacteria and Microalgae, Primary and Secondary (M ± SD).

Content, mg/g of Dry Biomass	*L. ectocarpi*	*N. shiloi*	*R. mixta* sp. nov.	*P. agardhii*	*T. viridis*
Proteins	560 ± 5	450 ± 5	680 ± 10	600 ± 10	350 ± 10
Lipids	250 ± 5	280 ± 5	50 ± 5	100 ± 5	195 ± 6
Carbohydrates	50 ± 3	70 ± 3	110 ± 5	180 ± 5	340 ± 5
FX	ND	15 ± 0.5	ND	ND	ND
C-phycoerythrin	78.8 ± 0.8	ND	35 ± 0.8	20 ± 1.2	ND
C-phycocyanin	22.0 ± 0.8	ND	140 ± 1	88.8 ± 1	ND
PUFA, include:	50 ± 1.2	67 ± 1.25	10 ±1.1	11 ± 1.2	70 ± 1.25
Eicosapentaenoic acid (C20:5 ω3)	ND	47.3 ± 0.61	ND	ND	4.8 ± 0.7
Arachidonic acid (C20:4ω6)	ND	8.5 ± 0.31	ND	ND	0.6 ± 0.6
γ-Linolenic acid (C18:3 n 6)	18 ± 0.04	0.22 ± 0.01	2.75 ± 0.02	1.5 ± 0.01	10.7 ± 0.05
Linoleic acid (C18:2 n 6)	25 ± 0.05	1.455 ± 0.04	3.16 ± 0.02	2.1 ± 0.01	2.5 ± 0.02
Oleic acid (C18:1 (n 9)	0.1 ± 0.02	1.46 ± 0.01	ND	2 ± 0.01	0,1 ± 0.02
α-Linolenic acid (C18:3 n 3)	10.0 ± 0.05	ND	ND	ND	6.5 ± 0.05

Note: FX, fucoxanthin; PUFA, polyunsaturated fatty acid; ND, not determined.

**Table 2 marinedrugs-21-00492-t002:** Effect of Diatomeae Fucoxanthin on In Vitro Bacterial Growth by MTT Assay (M ± SD).

Parameters	*K. pneumoniae* Clinical Isolate (s)	*K. pneumoniae* Clinical Isolate (r)	MSSA	MRSA	*S. aureus* Clinical Isolate (r)	*A. baumannii* Clinical Isolate (r)	*E. faecalis* Clinical Isolate (r)	*P. aeruginosa* Clinical Isolate (s)	*P. aeruginosa* Clinical Isolate (r)	*S. pyogenes* Clinical Isolate (s)
On Day 14										
Control	1.18 ± 0.08	1.75 ± 0.2	1.1 ± 0.09	1.47 ± 0.08	1.46 ± 0.16	0.98 ± 0.17	1.08 ± 0.09	1.94 ± 0.33	1.37 ± 0.33	1.88 ± 0.32
Cef	0.82 ± 0.08 *	1.24 ± 0.04 *	1.03 ± 0.14	1.32 ± 0.1 *	1.49 ± 0.13	0.7 ± 0.1 *	2.1 ± 0.32	1.29 ± 0.03 *	1.3 ± 0.03	1.53 ± 0.27
*A. platensis*	0.6 ± 0.17 *	0.8 ± 0.05 *#	0.83 ± 0.01 *#	0.96 ± 0.11 *#	0.96 ± 0.04 *#	0.68 ± 0.11 *	1.15 ± 0.13 #	0.49 ± 0.15 *#	0.67 ± 0.03 *#	1.3 ± 0.07 *
FX	0.83 ± 0.1 *	0.66 ± 0.05 *#	0.72 ± 0.08 *#	0.46 ± 0.07 *#	1.16 ± 0.05 *#	0.68 ± 0.14	2 ± 0.26	0.5 ± 0.06 *#	0.66 ± 0.07 *#	2.07 ± 0.29 #
On Day 21										
Control	2.52 ± 0.9	3.66 ± 0.39	4 ± 1	4 ± 0.9	4 ± 1	3.09 ± 1	4 ± 1	4 ± 0.8	4 ± 1	4 ± 0.9
Cef	2.15 ± 0.9	3.53 ± 0.54	2.07 ± 0.45 *	3.28 ± 0.96	1.23 ± 1.03 *	4 ± 0.9	4 ± 0.8	3.75 ± 0.51	3.23 ± 0.97	2.39 ± 1.67
*A. platensis*	0.54 ± 0.24 *#	2.33 ± 0.1 *#	1.98 ± 0.43 *	2.07 ± 0.51 *	2 ± 0.65 *	0.85 ± 0.29 *#	2.01 ± 0.65 *#	1.99 ± 0.69 *#	1.89 ± 0.79 *	2 ± 0.65 *
FX	1.11 ± 0.03 *#	2.75 ± 0.2 *#	2.26 ± 0.62 *	2.16 ± 0.92	2.68 ± 0.34 #	1.06 ± 0.01 *#	2.6 ± 0.34 *#	2.85 ± 0.2 *#	2.89 ± 0.1 *	2.2 ± 0.5 *

Note: Cef, ceftazidime; FX, fucoxanthin from *Nanofrustulum shiloi*; (s), sensitive strain; (r), resistant strain; * = *p* < 0.05 with control; # = *p* < 0.05 with ceftazidime. Ceftazidime was added to wells in doses of 50 µg/mL, FX in doses of 35 µg/mL, and microalgae extracts in doses of 1% *v*/*v*.

**Table 3 marinedrugs-21-00492-t003:** Cyanobacteria and Microalgae DMSO Extract’s Dose-Dependent Impact on Bacterial Growth as Measured by the MTT Assay (M ± SD).

Parameters	*N. shiloi*	*T. viridis*	*Pl. agardhii*	*L. ectocarpi*	*R. mixta* sp. nov.	FX
*K. pneumonia* clinical isolate (s) [Control = 1.38 ± 0.01, Ceftazidime = 0.92 ± 0.01]						
1% *v*/*v*	1.79 ± 0.22	1.7 ± 0.04	2.18 ± 0.01	2.06 ± 0.04	2.05 ± 0.04	1.4 ± 0.01
0.5% *v*/*v*	2.11 ± 0.08	2.01 ± 0.02	2.33 ± 0.02	1.91 ± 0.07	1,74 ± 0.05	1.35 ± 0.01
0.25% *v*/*v*	1.72 ± 0.01	1.75 ± 0.01	2.11 ± 0.08	1.8 ± 0.07	1.29 ± 0.01 *	1.18 ± 0.01 *
*K. pneumonia* clinical isolate (r) [Control = 1.83 ± 0.02, Ceftazidime = 1.72 ± 0.01]						
1% *v*/*v*	1.3 ± 0.01 *#	1.76 ± 0.01 *	1.7 ± 0.01	1.46 ± 0.01	1.36 ± 0.02 *#	1.18 ± 0.01 *#
0.5% *v*/*v*	1.19 ± 0.01 *#	1.68 ± 0. 01 *	1.68 ± 0.01 *	1.24 ± 0.02 *#	1.23 ± 0.01 *#	0.83 ± 0.01 *#
0.25% *v*/*v*	1.22 ± 0.01 *#	1.47 ± 0.01 *#	1.1 ± 0.01 *#	1.04 ± 0.03 *#	0.97 ± 0.01 *#	1.19 ± 0.01 *#
MSSA [Control = 1.92 ± 0.04, Ceftazidime = 0.43 ± 0.01]						
1% *v*/*v*	1.45 ± 0.01 *	1.19 ± 0.01 *	2 ± 0.01	1.53 ± 0.01 *	1.47 ± 0.01 *	1.61 ± 0.01 *
0.5% *v*/*v*	1.08 ± 0.01 *	1.27 ± 0.01 *	1.47 ± 0.01 *	1.69 ± 0.0 *	1.49 ± 0.01 *	1.53 ± 0.01 *
0.25% *v*/*v*	1.34 ± 0.01 *	1.43 ± 0.01 *	1.51 ± 0.01 *	1.38 ± 0.01 *	2.01 ± 0.01	1.19 ± 0.01 *
MRSA [Control = 2.11 ± 0.07, Ceftazidime = 2.34 ± 0.01]						
1% *v*/*v*	1.52 ± 0.03 *#	1.41 ± 0.01 *#	2.01 ± 0.02	1.92 ± 0.1 *#	2.16 ± 0.03	1.35 ± 0.01 *#
0.5% *v*/*v*	1.58 ± 0.01 *#	1.08 ± 0.01 *#	1.92 ± 0.01 *#	1.8 ± 0.01 *#	1.83 ± 0.01 *#	1.25 ± 0.01 *#
0.25% *v*/*v*	1.59 ± 0.03 *#	1.46 ± 0.01 *#	1.31 ± 0.01 *#	1.58 ± 0.01 *#	1.98 ± 0.01 *#	0.55 ± 0.01 *#
*A. baumannii* clinical isolate (r) [Control = 2.09 ± 0.03, Ceftazidime = 0.92 ± 0.01]						
1% *v*/*v*	2.92 ± 0.01	2.39 ± 0.02	3.62 ± 0.08	2.28 ± 0.01	2.23 ± 0.01	1.43 ± 0.02 *
0.5% *v*/*v*	2.75 ± 0.01	2.52 ± 0.12	2.68 ± 0.02	2.33 ± 0.04	2.02 ± 0.18	1.18 ± 0.24 *
0.25% *v*/*v*	2.15 ± 0.01	2.53 ± 0.06	2.64 ± 0.05	2.18 ± 0.01	2.03 ± 0.01	0.9 ± 0.1 *
*E. faecalis* clinical isolate (r) [Control = 2.61 ± 0.03, Ceftazidime = 0.92 ± 0.03]						
1% *v*/*v*	1.67 ± 0.01 *	1.54 ± 0.01 *	2.11 ± 0.01 *	2.16 ± 0.02 *	2.46 ± 0.02 *	1.08 ± 0.01 *
0.5% *v*/*v*	1.64 ± 0.01 *	1.82 ± 0.01 *	2 ± 0.01 *	2.18 ± 0.01 *	2.4 ± 0.01 *	1.01 ± 0.01 *
0.25% *v*/*v*	1.57 ± 0.01 *	1.56 ± 0. 01 *	1.38 ± 0.01 *	1.69 ± 0.03 *	1.61 ± 0.01 *	0.59 ± 0.01 *#
*P. aeruginosa clinical isolate* (*s*) [Control = 1.77 ± 0.01, Ceftazidime = 1.4 ± 0.05]						
1% *v*/*v*	2.8 ± 0.04	1.65 ± 0.01 *	2.83 ± 0.02	1.84 ± 0.01	1.55 ± 0.01 *	0.78 ± 0,01 *#
0.5% *v*/*v*	1.72 ± 0.01	1.45 ± 0.01 *	1.81 ± 0.01	1.68 ± 0.01	1.55 ± 0.01 *	0.55 ± 0.01 *#
0.25% *v*/*v*	1.5 ± 0.01	1.41 ± 0.01 *	1.47 ± 0.01 *	1.36 ± 0.01 *	1.59 ± 0.02 *	0.44 ± 0.01 #
*P. aeruginosa* clinical isolate (r) [Control = 1.73 ± 0.01, Ceftazidime = 1.25 ± 0.01]						
1% *v*/*v*	2.33 ± 0.02	1.35 ± 0.01 *	2.28 ± 0.02	1.46 ± 0.01 *	0.92 ± 0.01 *#	0.88 ± 0.01 *#
0.5% *v*/*v*	1.9 ± 0.01	2.13 ± 0.01	1,71 ± 0.01	1.36 ± 0.03 *	0.75 ± 0.01 *#	0.7 ± 0.01 *#
0.25% *v*/*v*	1.64 ± 0.01 *	1.45 ± 0.02 *	1.63 ± 0.01	1.19 ± 0.03 *	0.67 ± 0.01 *#	0.43 ± 0.01 *#
*S. aureus* clinical isolate (r) [Control = 0.98 ± 0.01, Ceftazidime = 0.99 ± 0.01]						
1% *v*/*v*	0.66 ± 0.01 *	0.84 ± 0.01 *	1.05 ± 0.01	0.91 ± 0.01 *	0.99 ± 0.03	0.47 ± 0.04 *#
0.5% *v*/*v*	0.53 ± 0.06 *	0.79 ± 0.03 *	0.95 ± 0.08	0.28 ± 0.02 *#	0.68 ± 0.02 *#	0.54 ± 0.04 *#
0.25% *v*/*v*	1.55 ± 0.07	0.36 ± 0.05 *	1.14 ± 0.06	0.99 ± 0.09	0.78 ± 0.08 *#	0.5 ± 0.02 *#

Note: FX, fucoxanthin from *Nanofrustulum shiloi*; (s), sensitive strain; (r), resistant strain; * = *p* < 0.05 with control; # = *p* < 0.05 with ceftazidime. Ceftazidime was added to wells in doses of 50 µg/mL, FX in doses of 35–17.5–8.75 µg/mL, and microalgae extracts in doses of 1% *v*/*v*.

**Table 4 marinedrugs-21-00492-t004:** Comparison of Cyanobacteria, Microalgae, and Chlorophyllipt DMSO Extracts Antimicrobial Activity (M ± SD).

Control	Ceftazidime	Chlorophyllipt	*N. shiloi*	*T. viridis*	*P. agardhii*	*L. ectocarpi*	*R.mixta* sp. nov.	FX
*K. pneumoniae* clinical isolate (r)								
2.27 ± 0.05	0.78 ± 0.01	0.98 ± 0.01 *	1.85 ± 0.01 *	1.53 ± 0.01 *	1.78 ± 0.01 *	1.93 ± 0.04 *	1.7 ± 0.01 *	0.77 ± 0.01 *&
MRSA								
3.02 ± 0.03	2.44 ± 0.01	0.72 ± 0.0 *#	0.9 ± 0.01 *#	1.2 ± 0.01 *#	1.93 ± 0.02 *#	0.93 ± 0.01 *#	0.9 ± 0.01 *#	0.6 ± 0.01 *#&
*A. bauman ii* clinical isolate (r)								
3.49 ± 0.03	0.92 ± 0.01	1.04 ± 0.01 *	1.6 ± 0.02 *	1.88 ± 0.04 *	1.86 ± 0.0 *	2.27 ± 0.02 *	1.53 ± 0.01 *	0.8 ± 0.01 *#
*E. faecalis* clinical isolate (r)								
2.29 ± 0.02	0.92 ± 0.01	0.57 ± 0.01 *#	1.41 ± 0.01 *	1.78 ± 0.01 *	1.82 ± 0.26 *	2.06 ± 0.01 *	1.7 ± 0.0 *	2.03 ± 0.02 *
*P. aeruginosa* clinical isolate (r)								
1.08 ± 0.01	0.93 ± 0.01	0.67 ± 0.01 *#	0.49 ± 0.01 *#&	1.23 ± 0.02	2.18 ± 0.03	2.54 ± 0.03	0.74 ± 0.001 *#	0.66 ± 0.01 *#

Note: FX, fucoxanthin from *Nanofrustulum shiloi*; (r), resistant strain; * = *p* < 0.05 with control; # = *p* < 0.05 with ceftazidime; & = *p* < 0.05 with chlorophyllipt. Ceftazidime was added to wells in doses of 50 µg/mL, FX in doses of 35 µg/mL, and microalgae extracts in doses of 1% *v*/*v*.

**Table 5 marinedrugs-21-00492-t005:** Cyanobacteria and Microalgae Extract’s Dose-Dependent Effect on Mycobacterium Growth In Vitro (M ± SD).

Parameters	*N. shiloi*	*T. viridis*	*P. agardhii*	*L. ectocarpi*	*R. mixta* sp. nov.	FX
	*Mycobacterium tuberculosis* strain H_37_Rv [Control = 1.38 ± 0.01, Rifampicin 100 µg/mL = 0.96 ± 0.04]					
1% *v*/*v*	1.74 ± 0.01	1.72 ± 0.01	1.33 ± 0.01 *	1.09 ± 0.01 *	0.62 ± 0.01 *#	1.18 ± 0.01 *
0.5% *v*/*v*	0.97 ± 0.01 *	1.41 ± 0.01	1.06 ± 0.01 *	0.89 ± 0.01 *#	0.79 ± 0.01 *#	0.73 ± 0.01 *#
0.25% *v*/*v*	0.67 ± 0.01 *#	1.18 ± 0.01	0.77 ± 0.01 *#	0.57 ± 0.01 *#	1.47 ± 0.01	2.06 ± 0.01
	*Mycobacterium smegmatis* [Control = 1.82 ± 0.02, Rifampicin = 1.6 ± 0.02]					
1% *v*/*v*	1.91 ± 0.01	1.08 ± 0.01 *#	2.76 ± 0.01	1.12 ± 0.01 *#	0.99 ± 0.01 *#	1.43 ± 0.01 *#
0.5% *v*/*v*	0.96 ± 0.01 *#	2.01 ± 0.01	0.93 ± 0.01 *#	1.01 ± 0.01 *#	0.76 ± 0.01 *#	1.13 ± 0.01 *#
0.25% *v*/*v*	1.46 ± 0.01 *#	1.94 ± 0.01	0.84 ± 0.01 *#	0.54 ± 0.01 *#	1.57 ± 0.01 *#	0.99 ± 0.01 *#

Note: FX, fucoxanthin *Nanofrustulum shiloi*; * = *p* < 0.05 with control; # = *p* < 0.05 with rifampicin. FX was added to wells in doses of 35–17.5–8.75 µg/mL.

## Data Availability

The data presented in this study are available upon request from the corresponding author.

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
