# Peer review of "Study of the Antimicrobial Potential of the Arthrospira platensis, Planktothrix agardhii, Leptolyngbya cf. ectocarpi, Roholtiella mixta nov., Tetraselmis viridis, and Nanofrustulum shiloi against Gram-Positive, Gram-Negative Bacteria, and Mycobacteria"

_marinedrugs, 2023, doi:10.3390/md21090492_

Round 1

Reviewer 1 Report

The paper of Lykov et al. reports on the activity of extracts from five among marine and freshwater microalgae against a panel of GRAM + and GRAM – bacteria and two Mycobacterium species, i.e. M. tuberculosis strain H37Rv and M. smegmatis.

The Introduction is quite confusing and misleading concepts are expressed: i.e. pigments, lipids, proteins and polysaccharides as such are primary metabolites but species/specific compounds belonging to these classes may show bioactivity in some biological assays. The authors even later on confound the definition of secondary and primary metabolites (i.e. L187). 

In addition, it’s true that some microalgal species produce toxins (i.e. dinoflagellates) but it is definitely false that only the species that the authors mentioned are less toxic and can be considered as ‘source of organic food, feed, nutraceuticals and pharmaceuticals’.

Results show biochemical data but no specific chemical analysis nor a chemical profile was reported in order to identify the molecules responsible of the putative antibacterial activity. Considering the level of scientific reports on bioactive compounds from microalgae with deep chemical investigation of extracts, fractions and pure compounds by using modern mass spectrometric and spectroscopic methodologies, it is nowadays no more acceptable such a general report based on extract activity. By the way, Mycobacterium spp are not fungi but bacteria thus it is wrong referring to anti-Mycobacterium as antimycotic activity.

A general antibacterial activity of microalgal extracts has been disclosed in many many papers. Hence, at this stage, I cannot see any element of novelty that can justify the publication on Marine Drugs.

There are several typos throughout the manuscript. Overall, an extensive editing of the English language in recommended.

Author Response

Response to Reviewer 1

Comments

We thanks the reviewer for working with us and will consider the comments made when revising the paper.

Point 1:

The Introduction is quite confusing and misleading concepts are expressed: i.e. pigments, lipids, proteins and polysaccharides as such are primary metabolites but species/specific compounds belonging to these classes may show bioactivity in some biological assays. The authors even later on confound the definition of secondary and primary metabolites (i.e. L187).

Response 1: With regard to the products of cellular metabolism in cells, there is no clear and unambiguous criterion for classifying them as primary or secondary metabolites, so there are two points of view on this problem: 1) according to some authors, primary metabolites include all metabolites formed in the process of normal cellular activity, including anabolites (proteins, polysaccharides, lipids, etc.) and catabolites (ethanol, organic acids, carbon dioxide, etc.). Secondary metabolites (idiolytes) include various specific compounds that can be formed in response to external factors that lead to the disruption of normal metabolic processes, such as nutrient deficiency, environmental pollution, contamination by toxic substances, etc. (2) Other authors believe that only protein is a secondary metabolite (idiolytes). 2) Other authors believe that only protein-enzymes can be called primary metabolites and that all other compounds are secondary metabolites - products catalysed by enzyme reactions. The first view is more widely accepted in biotechnology (Mitishev A.V., Kurdyukov E.E., Rodina O.P., Semenova E.F., Moiseeva I.Ya., Fadeeva T.M. Microalgae as a new source of biologically active compounds with antibacterial activity. Problems of biological, medical and pharmaceutical chemistry. 2021;24(7):24-29. https://doi.org/10.29296/25877313-2021-07-04). Perhaps the reviewer is referring to some generally accepted terminology; for example, when working with plants, all carotenoids of the photosynthetic apparatus are primary metabolites, and when the growth of green microalgae is restricted and at high irradiance, a large amount of secondary carotenoids accumulate in their cells as a protective response. This did not happen with our species of algae and cyanobacteria because we grew the cultures under non-limited conditions. You are right and we agree with your statement that proteins, fatty acids, vitamins and pigments should be identified as primary metabolites of microalgae. And secondary metabolites, such as substances that contribute to the defence of cells against stress conditions (temperature, light, etc.) (Yavuz Selim Cakmak, Murat Kaya, Meltem Asan-Ozusaglam. Biochemical composition and bioactivity screening of various extracts from Dunaliellasalina, a green microalga. EXCLI Journal. 2014; 679-690], inhibit the growth of various microorganisms should be considered such as cyanovirin, oleic acid, palmitoleic acid, vitamin E and B12, beta-carotene, phycocyanin, lutein and zeaxanthin [Michele Greque de Morais, Bruna da Silva Vaz, EtieleGreque de Morais, Jorge Alberto Vieira Costa. Biologically Active Metabolites Synthesised by Microalgae. BioMed Research Internationa. 2015; 2015:1-15. doi: 10.1155/2015/835761). It has been shown that antimicrobial action is predominantly possessed by representatives of the classes of green, diatom and golden algae, the level of synthesis of antimicrobial substances is influenced by water pollution (Falaise C., François C., Travers M.-A., Morga B., Haure J., Tremblay R., Mouget J.-L. Antimicrobial Compounds from Eukaryotic Microalgae against Human Pathogens and Diseases in Aquaculture. Mar. Drugs. 2016; 14(9): 159.)

Point 2: In addition, it’s true that some microalgal species produce toxins (i.e. dinoflagellates) but it is definitely false that only the species that the authors mentioned are less toxic and can be considered as ‘source of organic food, feed, nutraceuticals and pharmaceuticals’.

Response 2: You are right that other species of microalgae can be used for pharmacy, cosmetology and food supplements. Arthrospira (Spirulina) platensis, Chlorella or Chlorella vulgaris, Dunaliella, Aphanizomenon and Nostoc are authorised for human consumption [Santiago-Morales, I. S., Trujillo-Valle, L., Márquez-Rochoc. S., Trujillo-Valle, L., Márquez-Rocha, F. J., Hernández, J. F. L. (2018). Tocopherols, phycocyanin and superoxide dismutase from microalgae: As potential food antioxidants. Applied Food Biotechnology 5(1), 19-27. https://doi.org/10.22037/afb.v5i1.17884; Hu, J., Nagarajan, D., Zhang, Q., Chang, J.-S., Lee, D.-J. (2018). Heterotrophic cultivation of microalgae for pigment production A review. Biotechnology Advances, 36(1), 54-67. https://doi.org/10.1016/j.biotechadv.2017.09.009; Mazumdar, N., Novis, P. M., Visnovsky, G., Gostomski, P. A. (2019). Effect of nutrients on the growth of a new alpine strain of Haematococcus (Chlorophyceae) from New Zealand. Phycological Research, 67(1), 21-27. https://doi.org/10.1111/pre.12344]. The problem is that most of the microalgae studied do not have GRAS status, which limits their use as nutraceuticals.

Point 3: Results show biochemical data but no specific chemical analysis nor a chemical profile was reported in order to identify the molecules responsible of the putative antibacterial activity. Considering the level of scientific reports on bioactive compounds from microalgae with deep chemical investigation of extracts, fractions and pure compounds by using modern mass spectrometric and spectroscopic methodologies, it is nowadays no more acceptable such a general report based on extract activity.

Response 3: We agree with the reviewer's comment that the data provided on the biochemical composition of microalgae is insufficient and requires further research. In the first phase of the research, it was fundamental for us to obtain data on the antimicrobial potential of microalgae extracts, on the synergistic effect or lack thereof on microorganisms. Therefore, we used photosynthetic microorganisms from different systematic groups: cyanobacteria, green and diatom microalgae. In addition, marine and freshwater species were studied. These organisms, with their different ecological plasticity, are able to survive in extremely unfavourable habitat conditions. As a result, these organisms synthesise a number of different metabolites (identified and unknown to modern science) with different antibacterial activities. In the first phase of our research, we therefore carried out extensive prospecting studies to narrow down the search for valuable compounds. And then to isolate and identify the pure substances responsible for the manifestation of toxic effects on microorganisms.

Point 4: By the way, Mycobacterium spp are not fungi but bacteria thus it is wrong referring to anti-Mycobacterium as antimycotic activity.

Response 4: You are correct about the incorrect use of the term antifungal in relation to mycobacteria. This inaccuracy will be corrected.

Point 5: A general antibacterial activity of microalgal extracts has been disclosed in many many papers. Hence, at this stage, I cannot see any element of novelty that can justify the publication on Marine Drugs.

Response 5: We do not claim to be the first to investigate the antimicrobial potential of microalgae; our aim was to compare the antimicrobial potential of different microalgal taxa, including a new species of cyanobacterium that was obtained in 2021 and is still relatively unexplored. We also used different species of cyanobacteria and microalgae whose biomass we were able to obtain in an industrial photobioreactor. This is important because it is known that producers of high value that cannot be cultivated on an industrial scale are of little interest.

Reviewer 2 Report

Title: Antimicrobial Potential of the Microalgae Extracts

Summary: Microalgae are a source of carotenoids, phycocyanin, phenol, amino acids, polyunsaturated fatty acids, sulfated polysaccharides, pigments, and other bioactive molecules with antimicrobial, antioxidant, antiviral, antitumor, anti-inflammatory, and fat-burning properties. Thus, microalgae have moderate to weak antimicrobial activity. The article is well-constructed and supported by relevant studies. Following are my further suggestions for further improvements.

Review comments:

1.     Title: The title is too general and can be improved and written in a more specific way.

2.     Abstract: Need to be enhanced and written in a more scientific way in accordance with the results obtained. Include the novelty line and prospects line.

3.     Introduction: This section needs elaboration and needs to be improved with reference to relevant published kinds of literature. There is various literature available for the improvements.

4.     Results: Well, explained with efficient graphs and figures but extensive improvements are needed for the elaboration or to support the reproducibility.

5.     Line 62-66: Check the italics, some species are not properly abbreviated.

6.     Figure 1: Missing the T bars or error bars. (Standard Deviation)

7.     Discussion: Acceptable in the present format, some sections can be elaborated with the help of previously published literature.

8.     Material and methods: 4.1: Is too descriptive for methods and needs to summarize or support with relevant references.

9.  Conclusion: Too short, can be elaborated with the obtained and conclusive results with future prospects lines.

10.  Grammatical mistakes need to be considered. The species names must be in italics.

Grammatical mistakes need to be considered. The species names must be in italics.

Author Response

Response to Reviewer 2

Comments

We thanks the reviewer for working with us and will consider the comments made when revising the paper.

Point 1: Title: The title is too general and can be improved and written in a more specific way.

Response 1: We agree with the reviewer's recommendation and will change the title of the article.

Point 2: Abstract: Need to be enhanced and written in a more scientific way in accordance with the results obtained. Include the novelty line and prospects line.

Response 2: Thanks to the reviewer for the recommendation to improve the perception of the abstract.

Point 3: Introduction: This section needs elaboration and needs to be improved with reference to relevant published kinds of literature. There is various literature available for the improvements.

Response 3: We agree with the comment and will correct this section of the article.

Point 4: Results: Well, explained with efficient graphs and figures but extensive improvements are needed for the elaboration or to support the reproducibility.

Response 4: Thanks for the comment, will make adjustments.

Point 5: Line 62-66: Check the italics, some species are not properly abbreviated.

Response 5: Thanks for the comment, will make adjustments.

Point 6: Figure 1: Missing the T bars or error bars. (Standard Deviation)

Response 6: We apologise for the technical error that occurred when transferring the "Whiskers - Standard Deviation" graph from Excel. We're fixing it.

Point 7: Discussion: Acceptable in the present format, some sections can be elaborated with the help of previously published literature.

Response 7: Thanks for the comment, will make adjustments.

Point 8: Material and methods: 4.1: Is too descriptive for methods and needs to summarize or support with relevant references.

Response 8: Agree with the reviewer that this section of the Materials and Methods should be shortened.

Point 9: Conclusion: Too short, can be elaborated with the obtained and conclusive results with future prospects lines.

Response 9: Thanks for the good advice on correcting the conclusion, will be corrected.

Point 10: Grammatical mistakes need to be considered. The species names must be in italics

Response 10: We will take your comment into account and correct grammatical errors and spelling of microalgae taxa.

Reviewer 3 Report

Review Antimicrobial Potential of the Microalgae Extracts

The authors present a manuscript on crude extract of diverse microalgae and test those extracts in a variety of assays (toxicity, antimicrobial activity, etc.).

Unfortunately, the authors do not provide sufficient details of the algae used here (strain designation, 16S, etc.) for other labs to reproduce any of these results. Similarly, bacteria used in their antimicrobial assays are largely undefined without strain designation, 16S, etc. making the results of this manuscript very hard, if not impossible, to reproduce by other labs. This needs to be addressed.

The title of the manuscript is too broad and needs editing.

The authors extract fucoxanthin from algae and use olive oil as a solvent. The authors should use olive oil as negative control in their assays (e.g. fig 3 and fig 4). This needs to be addressed.

Try avoiding colloquial sentences like ‘However, not everything is so clear regarding the effectiveness of microalgae protection against bacteria’

English language is fine; minor editing required.

Author Response

Response to Reviewer 3

Comments

We thanks the reviewer for working with us and will consider the comments made when revising the paper.

Point 1: Unfortunately, the authors do not provide sufficient details of the algae used here (strain designation, 16S, etc.) for other labs to reproduce any of these results. Similarly, bacteria used in their antimicrobial assays are largely undefined without strain designation, 16S, etc. making the results of this manuscript very hard, if not impossible, to reproduce by other labs. This needs to be addressed.

Response 1:

Information on the genetics of 16S strains (Cyanobacteria):

Roholtiella mixta at Genbank: https://www.ncbi.nlm.nih.gov/nuccore/MK990636.1

Abdullin, S.R., Nikulin, V.Yu.; Nikulin, A.Yu.; Manyakhin, A.Yu.; Bagmet, V.B.; Suprun, A.R.; Gontcharov, A.A. Roholtiella mixtasp.nov. (Nostocales, Cyanobacteria): morphology, molecular phylogeny, and carotenoids content. Phycologia 2021, 60(1), 73-82. 466 doi: 10.1080/00318884.2020.1852846

Leptolyngbya cf. ectocarpi and  Planktothrix agardhii

The genetics have been done, but have not yet been added to the genebank. Here is a description of the collection site: Microphytobenthos samples were collected in 2020 from periphyton after the exposure of solid substrates in Karantinnaya Bay in the Black Sea (44°36′56″N, 33°30′10″E). To obtain a batch culture of periphyton cyanobacteria, a sample with a biofilm with an area of 1 cm2 was placed in a Petri dish with 30 mL of the liquid growth medium BG-11 prepared on sterile seawater (g/L): NaNO3 – 1.5; K2HPO4 × 3H2O – 0.04; MgSO4 × 7H2O – 0.075; CaCl2 × 2H2O – 0.036; citric acid – 0.006; ferric citrate – 0.006; Na2EDTA – 0.01; Na2CO3 – 0.02.  The culture was incubated under natural light at a temperature of 23 ± 2 °C for 2–4 weeks till the appearance of visible signs of growth.

Identified from the directory: Komárek, J., Anagnostidis K., 2005. Cyanoprokaryota. 2 Teil: Oscillatoriales. In: Büdel, G. Gärtner, L. Krienitz, M. (Eds.), Schagerl. Süßwasserflora von Mitteleuropa. Bd 19/2. München: Elsevier GmbH: 1-759.

Artrospira platensis at Genbank: https://www.ncbi.nlm.nih.gov/nuccore/MZ408912.1

Information on the genetics of 18S strains (Diatomaceae):

Nanofrustulum shiloi

The genetics have been saddled, added to GenBank (GenBank OR359397), but no number has yet been assigned.

Microbial Collection Information:

Museum strains of bacteria: Klebsiella pneumoniae (ATCC 13883™), Pseudomonas aeruginos (ATCC 27853), Staphylococcus aureus (ATCC® 25923™) and Staphylococcus aureus (ATCC BAA-1707).

Clinical isolates of Kl. pneumoniae, S. aureus, A. baumanii, E. faecalis and Ps. aeruginosa were isolated from pulmonary tuberculosis patients and verified by morphological and Mass spectrometry (MALDI TOF) matrix-assisted time-of-flight mass spectrometry (MALDI TOF) methods. As these are clinical isolates, it is not possible to reproduce them in other laboratories.

We specifically took clinical isolates of multi-drug resistant bacteria to test the antimicrobial effect of microalgae extracts not only on museum strains, but also on bacteria obtained from the Institute of Tuberculosis, which could a priori develop resistance to many antimicrobials due to the aggressive and prolonged chemotherapy given to patients.

Point 2: The title of the manuscript is too broad and needs editing.

Response 2: Thanks for the comment, we will take it into account in the new version of the article.

Point 3: The authors extract fucoxanthin from algae and use olive oil as a solvent. The authors should use olive oil as negative control in their assays (e.g. fig 3 and fig 4). This needs to be addressed.

Response 3:Response 4: Thanks for the valuable advice, will make changes.Thank you for pointing out our significant omission, will be corrected.

Point 4: Try avoiding colloquial sentences like ‘However, not everything is so clear regarding the effectiveness of microalgae protection against bacteria’

Round 2

Reviewer 1 Report

I do appreciate the efforts done by the authors while explaining the definition of secondary metabolites to this reviewer. However, the concept on the function of primary and secondary metabolism is largely consolidated for researchers working in the field. Among the others, it remains the fact quite disorienting that in Figure 1 "the chemical nature of the bioactive compounds in cyanobacteria and microalgae extract was analyzed by fermentative methods and found that in cyanobacteria and microalgae extracts present proteins, triglycerides and glucose": Which of these primary metabolites the authors associate to bioactivity? Bioactive, in which test or assay?

Then all the work is focused on a single metabolite, fucoxanthin, and the extracts and the fractions enriched in this metabolite, but there is no target quantitative assessment or specific chemical identification of this metabolite in any extract from cyanobacteria or microalgae. Any other metabolite could be responsible of the antimicrobial effect. Finally, pure fucoxanthin was tested in vivo, regardless from all that had (not) been demonstrated before. All the discussion is a quite boring report of literature data.

Moderate English style revision is required

Author Response

We sincerely appreciate the reviewer's hard work and extend our gratitude.

We were unable to properly explore the chemical makeup of cyanobacterial and microalgae extracts due to the limited human and technological capabilities. As a result, in the initial stage of the study, we restricted ourselves to using commercial kits for the immunoenzymatic method of determining the biochemical composition of human biological fluids in order to estimate the yield of primary metabolites in the extract. The following year will be dedicated to the purification of cyanobacterial and microalgae extract fractions. We make no assertions that the major metabolites (proteins, triglycerides, and sugars) we discovered are the bioactive substances with poisonous and antibacterial activities. We merely confirmed that these cyanobacteria and microalgae are diffusing metabolites into the eluent.

We describe the quantification in the alcoholic and oil extracts of fucoxanthin in the Materials and Methods section. The alcoholic extract contains fucoxanthin at a concentration of 0.35 mg/ml and the oil extract contains 0.5 mg/ml. You are quite right that other bioactive compounds such as lipids can diffuse into ethanol. We plan to obtain pure fucoxanthin in the future to screen out possible effects of other compounds. On the other hand, the combination of different metabolites enhances the antimicrobial potential of cyanobacterial and microalgae extracts as we have indicated in the article. The fact that fucoxanthin is used as a geroprotective, hepatoprotective drug in metabolic syndrome does not mean that it does not need to be investigated.

Reviewer 2 Report

The conclusion section needs to be improved. Not reached the journal quality and nor display the conclusive remarks of the full article.

Minor typos need to be corrected.

Author Response

We would like to express our deep and sincere gratitude for your consideration of our article.

Made changes to the end of the article, hoping for a positive response.

We deeply apologise for the typos in the article, we will correct them.

Reviewer 3 Report

Authors have addressed main concerns regarding source and details of species and strains. Manuscript needs minor English editing.

Minor edits recommended. New text has some typos here and there.

Author Response

We would like to express our deep and sincere gratitude for your consideration of our article.

We deeply apologise for the typos in the article, we will correct them.